# Preclinical safety study of a combined therapeutic bone wound dressing for osteoarticular regeneration

Laetitia Keller[1,2,11], Luc Pijnenburg[1,2,11], Ysia Idoux-Gillet[1,2,11], Fabien Bornert[1,2], Laila Benameur[3], Maryam Tabrizian [3,4], Pierrick Auvray[5], Philippe Rosset[6], Rosa María Gonzalo-Daganzo[7], Enrique Gómez Barrena[8], Luca Gentile [1,2,9,10] & Nadia Benkirane-Jessel[1,2]

The extended life expectancy and the raise of accidental trauma call for an increase of osteoarticular surgical procedures. Arthroplasty, the main clinical option to treat osteoarticular lesions, has limitations and drawbacks. In this manuscript, we test the preclinical safety of the innovative implant ARTiCAR for the treatment of osteoarticular lesions. Thanks to the combination of two advanced therapy medicinal products, a polymeric nanofibrous bone wound dressing and bone marrow-derived mesenchymal stem cells, the ARTiCAR promotes both subchondral bone and cartilage regeneration. In this work, the ARTiCAR shows 1) the feasibility in treating osteochondral defects in a large animal model, 2) the possibility to monitor non-invasively the healing process and 3) the overall safety in two animal models under GLP preclinical standards. Our data indicate the preclinical safety of ARTiCAR according to the international regulatory guidelines; the ARTiCAR could therefore undergo phase I clinical trial.

[1] INSERM (French National Institute of Health and Medical Research), UMR 1260, Regenerative Nanomedicine (RNM), FMTS, 11 Rue Humann, 67000 Strasbourg, France. [2] Université de Strasbourg, Faculté de chirurgie dentaire, Hôpitaux Universitaires de Strasbourg, 8 Rue de Ste Elisabeth, 67000 Strasbourg, France. [3] Department of Biomedical Engineering, McGill University, 3775 Rue University, Montréal, QC H3A24, Canada. [4] Faculty of Dentistry, McGill University, 2001 Avenue McGill, Montréal H3A1G1, Canada. [5] C.Ris Pharma, Parc Technopolitain - Atalante Saint-Malo, 35400 Saint Malo, France. [6] Faculté de Médecine de Tours, CHRU de Tours, Service de Chirurgie Orthopédique 2, Université François Rabeleis, 37044 cedex 9 Tours, France. [7] Unidad de Producción Celular, S° de Hematología y Hemoterapia, Hospital Universitario Puerta de Hierro Majadahonda, 28222 Majadahonda, Madrid, Spain. [8] Orthopaedic Surgery and Traumatology Department, Hospital La Paz-IdiPAZ and Universidad Autónoma de Madrid, E-28046 Madrid, Spain. [9] Hasselt University, Campus Diepenbeek, Agoralaan, Gebouw D, 3590 Diepenbeek, Belgium. [10] University of Applied Sciences, Kaiserslauter, Campus Zweibrücken, Amerikastr. 1, 66482 Zweibrücken, Germany. [11]These authors contributed equally: Laetitia Keller, Luc Pijnenburg, Ysia Idoux-Gillet. Correspondence and requests for materials should be addressed to N.B.-J. (email: nadia.jessel@inserm.fr)

Regeneration of osteochondral defects represents a major challenge, especially considering the aging of the population. The surgical procedures currently applied (bone graft, mosaicplasty, micro-fracture, articular prosthesis, therapeutic implant), are invasive and/or painful for the patient, with limited efficacy and side effects. Lesions of the femoral condyles, for example, are a common side effect that could have serious consequences[1]. A 2002 study found that ≥60% of patients undergoing arthroscopy showed osteochondral defects[2]; in more than half of the cases, such a lesion was classified as grade 3 or higher, according to the International Cartilage Repair Society (ICRS) scale. Osteochondral defects do not heal properly and, even when treated (e.g. by Pridie's marrow stimulation or by mosaicplasty treatment) consistently led to osteoarthritis (OA)[3,4]. This inevitably has a high impact on the public health system, with the direct costs of the treatment, but it also has repercussions on the general economy (social costs and loss of economic production), setting the overall costs of the disease between 0.25 and 1% of a country's GDP[5,6].

The unique properties of the cartilage (multilayered cell structure, different extracellular matrix composition and fibril orientation) make it difficult to repair. Surgical techniques like micro-fracture, mosaicplasty, osteoarticular transplantation or autologous chondrocytes implant may allow a partial functional recovery, but are mostly aimed to relieve the pain and prevent the lesion to spread[7]. Besides their variable outcome[8] and intrinsic limitations[9,10] none of the afore-mentioned techniques was shown to restore the hyaline articular surface[11], justifying the search for alternatives to promote osteoarticular regeneration (OAR). Recently, autologous chondrocytes pre-cultured on a membrane of mammalian collagen[12–15], were used to fill articular focal lesions and promote cartilage regeneration. However, when performed on subchondral bone, they showed site morbidity and fibrocartilage formation[16], leading to a dysfunctional repair. To overcome these limitations, mesenchymal stem cell (MSCs)-based therapies emerged, which employ autologous bone marrow-derived MSCs to increase the efficiency of OAR[17–20]. A combination of biomaterials, stem cells and active molecules are therefore needed to promote an effective tissue repair and to achieve a functional recovery of the articulation[21,22].

Recently, we proposed the ARTiCAR (ARTicular CArtilage and subchondRal bone implant) combined Advanced Therapy Medicinal Products (ATMPs) for personalized OAR (Fig. 1a, b). The implant is made of a nanofibrous wound dressing (FDA-approved resorbable polymeric Poly-ε-caprolactone)[23–27] and autologous bone marrow-derived MSCs[27–29] (Fig. 1a). The wound dressing is nanofunctionalized with nanoreservoirs, for cell contact-dependent delivery of physiological concentrations of bone morphogenetic factor 2 (BMP2). The nanoreservoirs technology enabled to reduce the dose of BMP2 to physiological levels, making it locally and sustainably available, and reducing the adverse effects of its massive release, e.g. from soaked collagen sponges currently used in the clinic[30–32].

In this work, we test the safety of the ARTiCAR combined ATMPs in two different animal models. Moreover, we assess the feasibility of non-invasive monitoring of the healing process in sheep, via MRI. The results of the toxicity and biodistribution tests, run accordingly to the international regulatory guidelines for cell therapies and medical devices[33–37] and Good Laboratory Practice (GLP)[38], prove the biosafety of the ARTiCAR combined ATMPs, which can therefore be used in phase I clinical trials as a ready-to-use, flexible implant to address both cartilage and subchondral bone regeneration in OA patients.

## Results

### Cytotoxicity of the NanoM1-BMP2 wound dressing in vitro.
The nanofibrous PCL wound dressing component (NanoM1-BMP2; Fig. 1a) of the ARTiCAR combined ATMPs was tested for cytotoxicity on MRC-5 fetal lung fibroblasts in vitro. Cells were seeded in the presence of the NanoM1-BMP2 wound dressing and compared to positive (polyurethane film; RM-A) and negative (high density polyethylene film; RM-C) controls. Different sizes of both the NanoM1-BMP2 membrane and the control films were tested in the range of 1-20 mm². Cell density and morphology were qualitatively evaluated by bright field microscopy. Cells cultured in the presence of RM-A started to detach already after 24 h (Fig. 2a–e). One the contrary, cells cultured in the presence of the NanoM1-BMP2 scaffold did not show any morphological abnormalities (Fig. 2f–j), as they did those cultured in the presence of RM-C films (Fig. 2k–o). Next, we assessed the viability of the MRC-5 cells in the 3 conditions tested, using the WST-1 live/dead cell assay. Both RM-A and NanoM1-BMP2 showed a decrease in cell viability over 24 h that was directly proportional to the size of the membrane used, as the interpolated trend lines indicated (solid black lines in Fig. 2p–r). However, in the presence of 20 mm² RM-A, the cell viability reduced to 72 ± 5% compared to t0 (Fig. 2p, $p \leq 0.05$), while in the presence of a fragment of NanoM1-BMP2 of the same size, the cell viability reduced to 97 ± 5% (Fig. 2q). No significant reduction of the cell number was also observed in the presence of the negative control film (Fig. 2r). These results indicate that the NanoM1-BMP2 is not toxic to MRC-5 cells in vitro.

### Acute toxicity after the implant of the ARTiCAR in nude rats.
The acute toxicity of the ARTiCAR was evaluated in vivo in nude rats, and compared to the non active part of the implant (hydrogel without hMSCs) as a vehicle. Clinical, hematological and biochemical parameters were evaluated. The biodistribution and the persistence of the transplanted cells were also assessed. Briefly, ARTiCAR combined ATMPs (group 1) or vehicles (group 2) were implanted into femoral defects in nude rats. Ventricular blood was taken before the animals were euthanised, 7 days post implant, and femurs were collected for histopathology analysis. Neither the ARTiCAR combined ATMPs, nor the vehicle triggered any significant effect on the body weight, either in female or male rats, over a period on 90 days following the implant (Fig. 3a). Hematological parameters (Fig. 3b) showed no significant differences among the 4 groups of animals (group 1 male, group 1 female, group 2 male, group 2 female). Biochemical parameters were also assessed (Fig. 3c). Female rats in Group 1 showed significantly higher plasmatic concentrations of both Alanine aminotransferase (ALAT; 17.0 ± 2.0 U/l vs. 13.2 ± 1.6 U/l for group 1 and 2, respectively; $p \leq 0.05$) and Calcium (91.74 ± 1.02 mg/l vs. 88.46 ± 0.91 mg/l for group 1 and 2, respectively; $p \leq 0.05$) than those in group 2. These differences were not associated with any additional symptoms and, altogether, the analysis of the hematological and biochemical parameters considered did not show any clinically relevant differences between ARTiCAR-treated animals and the control group. The femur-tibia joints were also collected and subject to histological analysis. Both the ARTiCAR and the vehicle induced comparable levels of inflammatory response at the implant site (delimited by asterisks in 3d, e), compatible with the bone healing process of the induced bone defect (Fig. 3d, e). Eventually, the biodistribution of the human MSCs at day 90 post implant was also assessed, using qPCR for detecting human DNA. Signal from hMSCs DNA was never detected above the threshold level, except in the testis of one male rat in group 1. The migration of the PCR product on 2.5% agarose gel confirmed the specificity of the amplification product. Taken together, clinical, hematological and biochemical data suggest that the ARTiCAR implant did not induce any clinically relevant symptoms; the inflammatory response detected from the histological analysis of the implant site revealed no differences

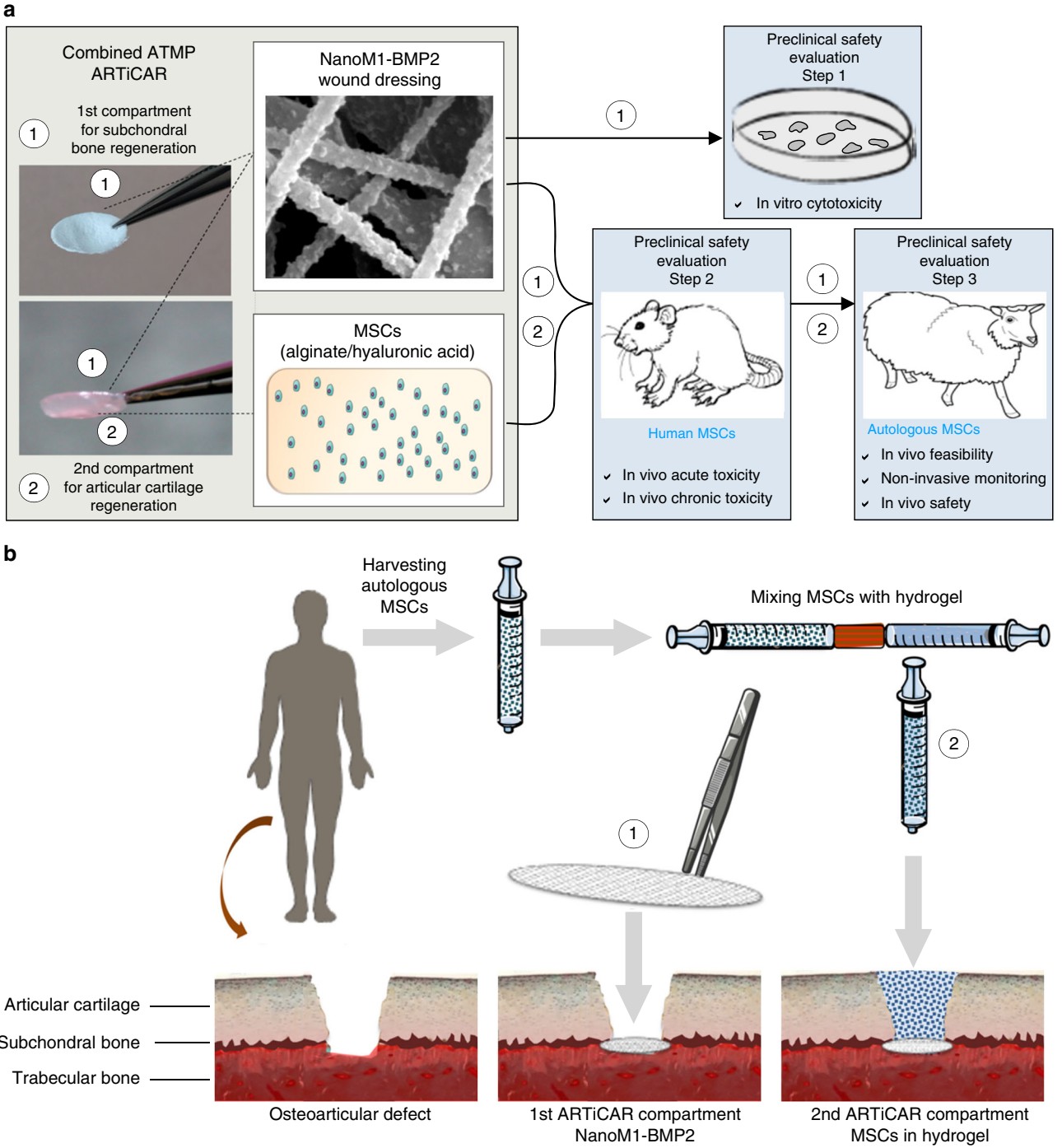

**Fig. 1** Composite advanced therapy medicinal product (ATMP) developed for the osteoarticular regeneration (OAR) of cartilage focal lesions. **a** The ARTiCAR is a composite ATMP that combines an FDA-approved synthetic wound dressing and a living therapeutic made of autologous mesenchymal stem cells (MSCs), embedded in alginate/hyaluronic acid hydrogel. The wound dressing, named NanoM1-BMP2 is made of nanofibrous poly-ε-caprolactone nano-functionalized with Bone morphogenetic protein 2 and aims at subchondral bone regeneration. The cellular hydrogel aims at articular cartilage regeneration. The NanoM1-BMP2 was tested for cytotoxicity in vitro (step 1); the whole ARTiCAR was tested for acute toxicity, biodistribution and persistence in an osteochondral defect nude rat model (step 2) and for feasibility, safety, and non-invasive monitoring of the procedure (step 3) in a sheep intra-articular model. **b** The ARTiCAR aims at the simultaneous regeneration of both the articular cartilage and the subchondral bone in a one-step surgical procedure. After harvesting MSCs from patient's bone marrow, the NanoM1-BMP2 is applied in contact with the injured subchondral bone; afterwards, the MSCs mixed with the hydrogel is applied to fill the defect

with the control group, strongly indicating the safety of the ARTiCAR implant for the treatment of bone defects.

**Intra-articular implant of the ARTiCAR in sheep**. To further confirm the safety of the ARTiCAR combined ATMPs, and to assess the feasibility of its usage in large animals, osteoarticular defects were induced in femoral condyles of adult sheep and were either left unfilled (no-treatment control: NT) of filled with the ARTiCAR implant or with an autograft (AG), as for supplemental table 1. The healing process was monitored non-invasively by

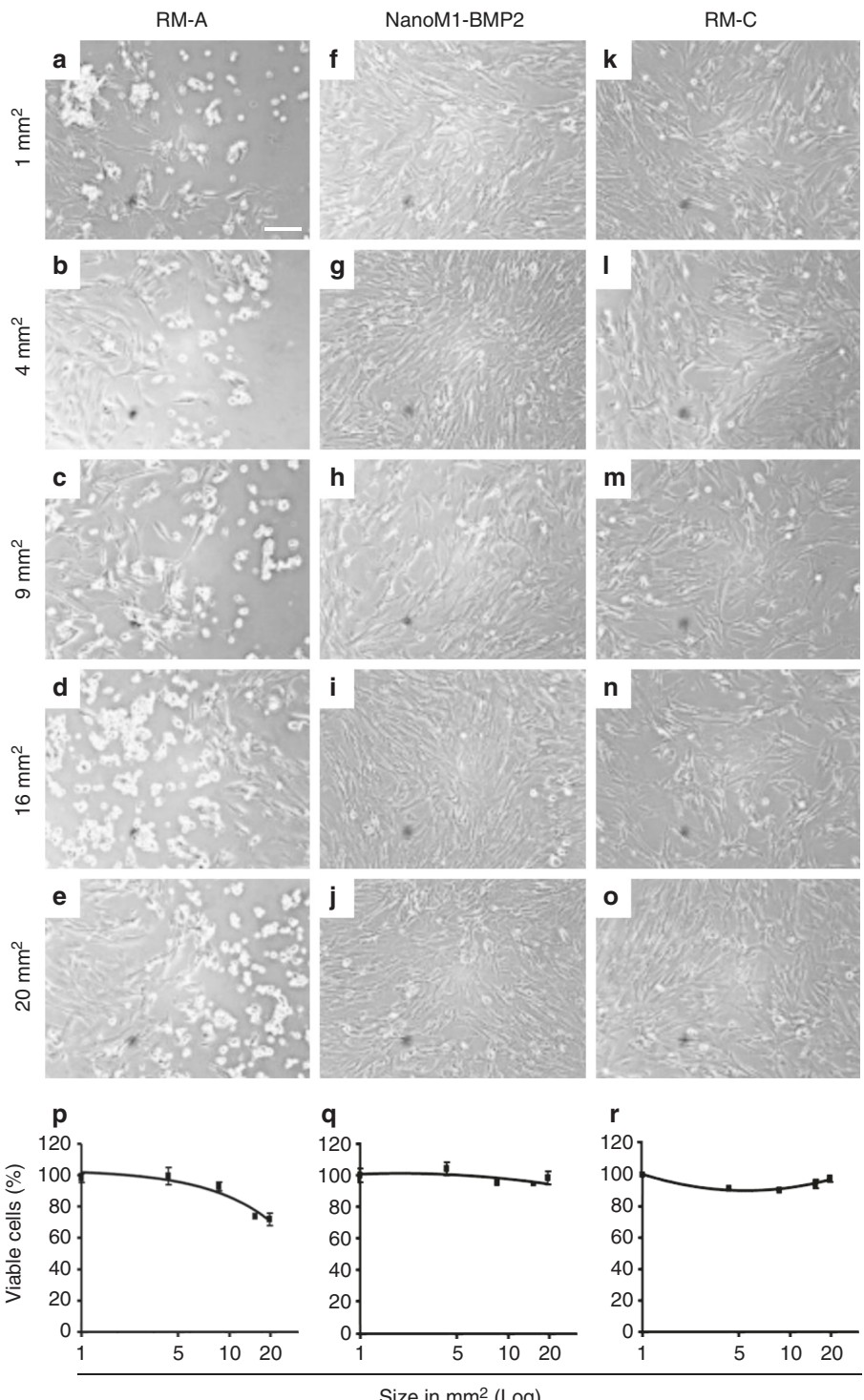

**Fig. 2** In vitro evaluation of the NanoM1-BMP2 wound dressing cytotoxicity. MRC-5 cell line was cultured in the presence of Polyurethane film, (RM-A; **a**–**e**), NanoM1-BMP2 wound dressing (**f**–**j**) or high density polyethylene film (RM-C; **k**–**q**). Five sizes (20, 16, 9, 4, 1 mm²) of membranes were used. Cells were inspected for morphological changes over a 3 day period. Proliferation/viability was assessed via WST-1 assay (**p**–**n**) at 24 h (C.Ris Pharma, CRO, France). Error bars represent standard deviation. Scale bar: 100 μm

means of magnetic resonance imaging (MRI) at 0, 12 and 26 weeks (Fig. 4a–c). After either 12 or 26 weeks from implant, sheep were euthanized and the femur-tibia joints were explanted and scanned via micro-computing tomography (micro-CT; Fig. 4d); a 3D surface rendering of the joint was also built from 2D section images (Fig. 4e). The explanted joints were macroscopically scored according to the ICRS score system as follows:

grade I = normal cartilage, grade II = nearly normal, grade III = abnormal cartilage and grade IV = severely abnormal cartilage (Fig. 4f). Eventually, the explants were stained in a solution of safranin o–fast green and examined histologically (Fig. 4g). The following parameters were taken into consideration within the repaired tissue and scored according to ICRS II score system[39]: subchondral bone abnormalities/marrow fibrosis, tissue

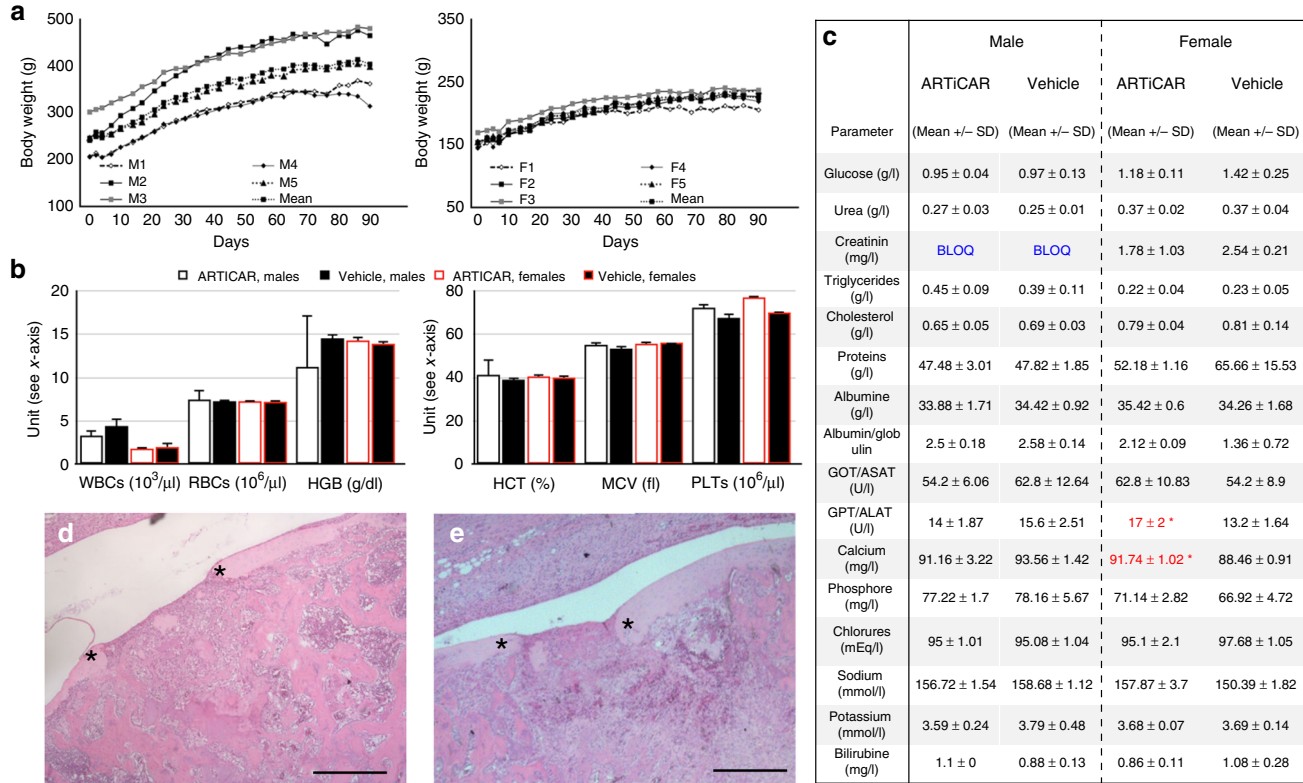

| c | Male | | Female | |
|---|---|---|---|---|
| | ARTiCAR | Vehicle | ARTiCAR | Vehicle |
| Parameter | (Mean +/- SD) | (Mean +/- SD) | (Mean +/- SD) | (Mean +/- SD) |
| Glucose (g/l) | 0.95 ± 0.04 | 0.97 ± 0.13 | 1.18 ± 0.11 | 1.42 ± 0.25 |
| Urea (g/l) | 0.27 ± 0.03 | 0.25 ± 0.01 | 0.37 ± 0.02 | 0.37 ± 0.04 |
| Creatinin (mg/l) | BLOQ | BLOQ | 1.78 ± 1.03 | 2.54 ± 0.21 |
| Triglycerides (g/l) | 0.45 ± 0.09 | 0.39 ± 0.11 | 0.22 ± 0.04 | 0.23 ± 0.05 |
| Cholesterol (g/l) | 0.65 ± 0.05 | 0.69 ± 0.03 | 0.79 ± 0.04 | 0.81 ± 0.14 |
| Proteins (g/l) | 47.48 ± 3.01 | 47.82 ± 1.85 | 52.18 ± 1.16 | 65.66 ± 15.53 |
| Albumine (g/l) | 33.88 ± 1.71 | 34.42 ± 0.92 | 35.42 ± 0.6 | 34.26 ± 1.68 |
| Albumin/globulin | 2.5 ± 0.18 | 2.58 ± 0.14 | 2.12 ± 0.09 | 1.36 ± 0.72 |
| GOT/ASAT (U/l) | 54.2 ± 6.06 | 62.8 ± 12.64 | 62.8 ± 10.83 | 54.2 ± 8.9 |
| GPT/ALAT (U/l) | 14 ± 1.87 | 15.6 ± 2.51 | 17 ± 2 * | 13.2 ± 1.64 |
| Calcium (mg/l) | 91.16 ± 3.22 | 93.56 ± 1.42 | 91.74 ± 1.02 * | 88.46 ± 0.91 |
| Phosphore (mg/l) | 77.22 ± 1.7 | 78.16 ± 5.67 | 71.14 ± 2.82 | 66.92 ± 4.72 |
| Chlorures (mEq/l) | 95 ± 1.01 | 95.08 ± 1.04 | 95.1 ± 2.1 | 97.68 ± 1.05 |
| Sodium (mmol/l) | 156.72 ± 1.54 | 158.68 ± 1.12 | 157.87 ± 3.7 | 150.39 ± 1.82 |
| Potassium (mmol/l) | 3.59 ± 0.24 | 3.79 ± 0.48 | 3.68 ± 0.07 | 3.69 ± 0.14 |
| Bilirubine (mg/l) | 1.1 ± 0 | 0.88 ± 0.13 | 0.86 ± 0.11 | 1.08 ± 0.28 |

**Fig. 3** clinical, hematological, biochemical and histopathology evaluation of the ARTiCAR safety in an osteochondral defect nude rat model. **a** Individual body weight curves of five male (M1–M5) and five female (F1–F5) rats, after implant of the ARTiCAR ATMP. **b**, **c** Seven days after implantation, ventricular blood of implanted rats was collected (group 1: ARTiCAR, group 2: hydrogel without hMSCs; ($n = 20$) and analyzed for hematological and biochemical parameters. WBCs: white blood cells; RBCs: red blood cells; HGB: haemoglobin; HCT: hematocrits; MCV: mean corpuscular volume; PLT: platelets. Data are presented as the mean ± SD. One-way ANOVA followed by the Bonferroni post-hoc test was performed for significance. A $p$ value ≤ 0.05 was considered significant. (C.Ris Pharma, CRO, France). **d**, **e** Histological sections of the intra-articular implant site in nude rats with femoral bone/cartilage defect, 7 days after surgery ($n = 20$) for group 1 (**d**) and 2 (**e**), at ×5 magnification. Comparable inflammatory responses involved in bone healing process and no systemic changes were observed in both group. Asterisks represent the limits of the defect. Scale bar: 1 mm

morphology, cell morphology, basal integration, formation of a tidemark, vascularization, overall assessment and mid/deep zone assessment (supplemental table 2). In general, a correct subchondral bone formation, a proper osteochondral remodeling zone and a good integration between graft and host tissues were observed within the induced bone defect 26 weeks after treatment in all the experimental groups considered (Fig. 4g, h). Interestingly, 12 weeks post implant, the ARTiCAR-treated defects showed reduced vascularization (66 ± 16%), poorer quality of the subchondral bone (25 ± 24%) and reduced tidemark (29 ± 34%) (Fig. 4h, left panels). These scores were aligned to the NT group (80 ± 14%; 54 ± 30%; 44 ± 30%, respectively), but significantly different to the AG group (100 ± 0%, $p ≤ 0.05$; 88 ± 6%, $p ≤ 0.05$; 80 ± 16%; $p ≤ 0.1$, respectively). However, the ARTiCAR group showed a better vascularization (2.5 ± 0.5) and a higher degree of fibrosis (2.75 ± 0.5) then the AG groups in the tissues adjacent to the defect ($p ≤ 0.1$ for both parameters), but not a level of fibrocartilage formation as high as in the NT control ($p ≤ 0.1$ and $p ≤ 0.05$ at 12 and 26 weeks post implant, respectively) (Fig. 4h, right panels, black arrow in Fig. 4j). These results suggest that the ARTiCAR might induce a low inflammatory response that in turn triggers regeneration[40], inducing subchondral bone formation (yellow arrow in Fig. 4k–n) and promoting effective healing in the long term. Moreover, at 26 weeks post implant, matrix staining, surface/superficial assessment, mid/deep zone assessment and overall assessment showed the highest scores for the ARTiCAR group (supplemental table 2), and no polymorphonuclear cells,

infection, fibrinous exudates and fatty infiltrates were detected in the tissue adjacent to the defect in both the ARTiCAR and the AG groups (supplemental table 3), indicating that the ARTiCAR combined ATMPs treatment has a safety over the long term comparable to that of an autograft.

## Discussion

The global cartilage repair/regeneration market is valued at USD 4.2 billion in 2016 and is expected to grow at a CAGR of 5.4% during the 2014-2025 period, owing to the increased life expectancy. Currently, the international standard treatment for OA is total knee arthroplasty (TKA). Despite its rapidly increasing utilization (77,000 patients/year younger than 55 years, in the US), TKA is prone to complications, like a higher risk of infection, persistent knee pain, patellar resurfacing problems and prosthetic fracture[41]. Once complications developed, the consequences for the patients are severe, which is why 20% of the patients who underwent TKA were unsatisfied abut the surgery outcome[42]. In most cases, revisions were required within 2-5 years after the primary implant[43]. Regenerative nanomedicine combines the use of biomaterials, nanotechnologies and cells to offer better solutions to issues like OAR, where a complex interface regeneration is required. In this work, we assessed the feasibility, non-invasive monitoring and safety of the ARTiCAR combined ATMPs. Similarly to other smart implantable scaffolds that promote osteochondral differentiation[44–46], the ARTiCAR releases a bone promoting factor. However, thanks to our

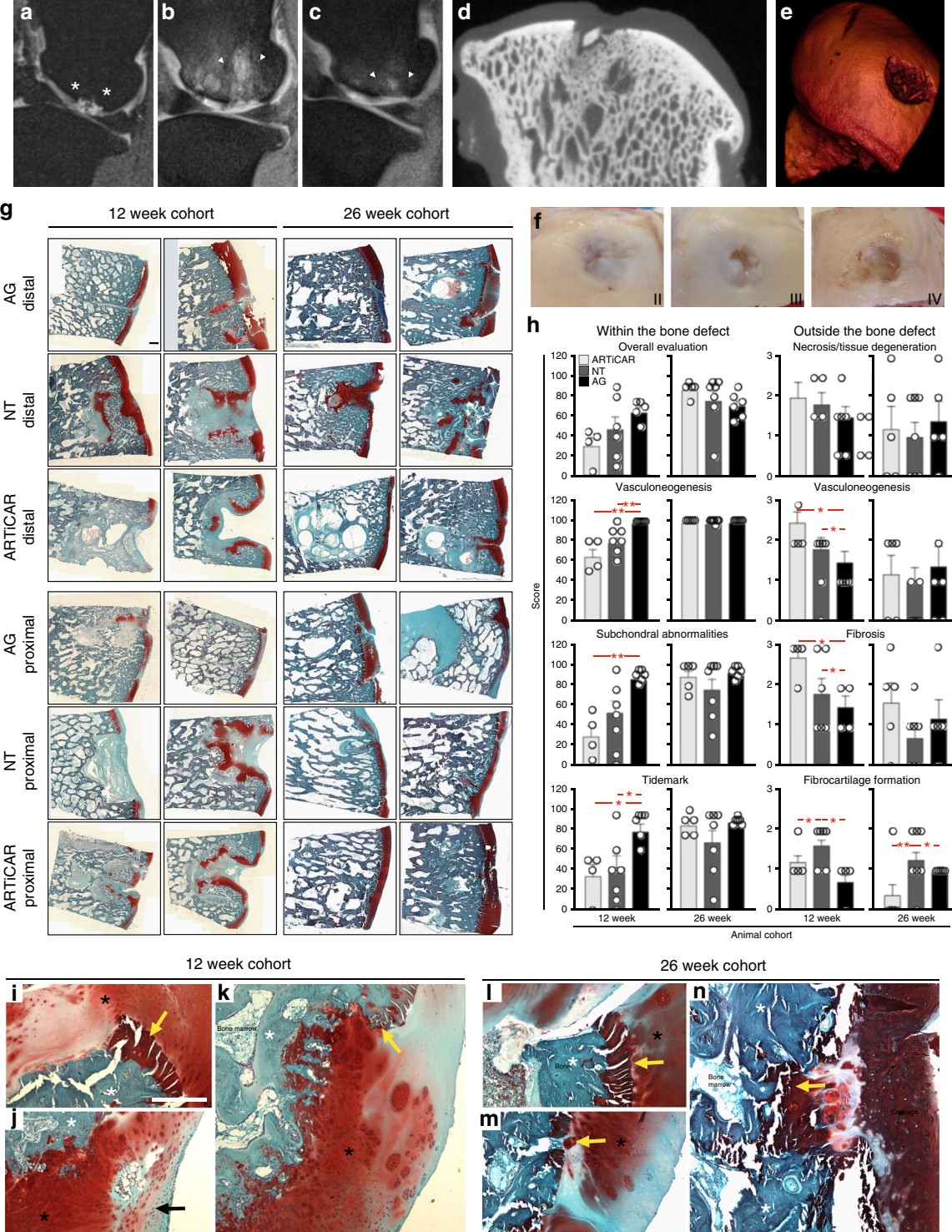

**Fig. 4** Feasibility, non-invasive monitoring and safety evaluation of the ARTiCAR combined ATMPs implanted in sheep intra-articular defect model. **a–e** the OAR process after ARTiCAR implant was monitored non-invasively by means of MRI immediately after surgery (**a**) and at 12 (**b**) and 26 (**c**) weeks post implant. The extension of the bone defect (white asterisks) and the ongoing OAR (white arrowheads) are visible. Micro CT-scan (**d**) followed by 3D surface rendering (**e**) were performed on freshly explanted joints. **f** The level of cartilage regeneration was macroscopically assessed based on ICSR score system. **g**, **h** Whole explants were sectioned and stained with safranin o - fast green (blue: bone; red: cartilage), imaged at ×4 and stitched together (**g**) to evaluate the OAR according to the ICRS II score system (**g**). Scale bar: 1 mm. ICRS II parameters were examined for ARTiCAR and the control groups considered and analyzed by mean of two-way ANOVA followed by Bonferroni post hoc test (**h**). * = $p \leq 0.1$; ** = $p \leq 0.05$. Dots represent individual scores, bars represent mean scores and error bars represent standard errors. (**i–n**) Zones of osteochondral remodeling within the treated defects in animals from AG (I,L), NT (**j**, **m**) and ARTiCAR (**k**, **n**) groups, either at 12 or 26 weeks post implant. Bone (white asterisks), cartilage (black asterisk), fibrocartilage (black arrow) and subchondral bone (yellow arrow) are shown. Scale bar: 500 μm

patented nanoreservoir technology that provide cell contact-dependent gradual release, the total amount of BMP2 used in the ARTiCAR is 10.000 times lower than that of BMP2-soaked collagen membranes used in the clinic[30], reducing both potential inflammatory side effects (Fig. 3d; Fig. 4g, h, k, n) and the overall costs of the procedure. Differently to other approaches where a poor subchondral bone regeneration was achieved[47,48], the ARTiCAR address simultaneous regeneration of both the subchondral bone and the cartilage[24–27,29] (Fig. 4g, k, n), representing an innovative technology for promoting OAR in a localized osteochondral defect. For cartilage regeneration, the ARTiCAR incorporates MSCs. Human MSCs are currently used in clinical trials for promoting OAR[49–51], because of their trans-differentiation potential coupled to immunomodulatory effect[52,53]. In future clinical trials, the MSCs could be directly harvested from patients, allowing for autologous transplantation. In order to be transplanted in human, MSCs will undergo an implemented quality control (supplemental table 4), and will be used only if the minimum release criteria for this type of ATMP are satisfied[54]. However, since tumorigenicity of MSCs is still debated[55,56], the biodistribution of hMSCs is a critical concern of preclinical safety[57,58]. After the implant of ARTiCAR, traces of hMSC DNA were found in the testes of one male nude rat, out of 40 implanted animals (Fig. 3c). Also, no tumor formation was observed in the transplanted rats and sheep, neither at the implant site, nor in the rest of the body. Altogether, these data pinpoint the safety of the ARTiCAR implant in respect to the tumorigenicity of the transplanted MSCs.

In summary, we showed the feasibility of the ARTiCAR implant in a large animal model and the possibility to follow OAR non-invasively, by mean of MRI (Fig. 4a–c). More importantly, we showed the safety of the ARTiCAR, as no acute or long term toxicity was detected, neither in nude rats (Fig. 3d), nor in sheep (Fig. 4g, h, k, n). Therefore, the ARTiCAR can enter phase I clinical trials as a treatment for osteochondral defects, with the potential to be used also for other conditions, like tendon degeneration and age-related degenerative musculoskeletal issues. As such, the ARTiCAR could replace more invasive current treatments, with the potential to impact 300.000 to 450.000 patients/year only in the US ($4-5 billion global market).

## Methods

**Study design**. All the experiments in this study were planned and performed according to the international regulatory guidelines[33–37] for cell therapies and medical devices. Good Laboratory Practice[38] and Standard Operating Procedures (SOPs) for all protocols were used. In vitro cytotoxicity assay was done according to ISO 10993-5 (2009 and 2012) guidelines. Assessment of the OAR was done in accordance to the ICRS II score system[39].

**Production of the nanofibrous compartment of the ARTiCAR**. The nanofibrous component of ARTiCAR was obtained via electrospinning of PCL, as previously described[24]. Briefly, PCL (PURASORB®, PURAC, Corbion, Amsterdam, Netherlands) was dissolved in a 25% (wt/vol) dimethylformamide/dichloromethane solution (3/2, v/v) and delivered at a constant rate of 1 ml/h to the EC-DIG electrospinning device (IME Technologies, Eindhoven, Netherlands), set to at high voltage (20 ± 3 kV). Following electrospinning, PCL membranes were kept in a desiccator at 45 °C, to remove residual solvents, and sterilized by gamma irradiations (25 kGy). Membranes were then dipped alternately in 200 µg/ml BMP2 solution (rh-BMP2, Inductos, Medtronic, France) in 40 mM 4-Morpholinoethanesulfonic acid (Sigma-Aldrich, Saint-Quentin Fallavier, France), 150 mM Sodium Chloride (Sigma-Aldrich), pH 5.5 (MES buffer) and 0.5 mg/ml Chitosan (Protasan UP CL 113, Novamatrix, Sandvika, Norway), for 12 times. Each bath was followed by three washes in MES buffer.

**Production of the hydrogel compartment of the ARTiCAR**. Twelve mg/ml sodium alginate (Sigma-Aldrich) and 3 mg/ml hyaluronic acid (Lifecore Biomedical, Chaska, USA) were dissolved in 9 mg/ml Sodium Chloride (Sigma-Aldrich). Prior to implant, the hydrogel was mixed with either human or sheep MSCs. After the MSC/hydrogel compartment was applied to fill the defect, gelation was achieved using 102 mM calcium chloride (Sigma-Aldrich).

**Cell culture**. Human lung fetal fibroblast MRC-5 cell line (ECACC, Sigma-Aldrich) was cultured in 75 cm²-flasks with EMEM (Lonza, Levallois-Perret, France) containing 10% Fetal Bovine Serum (Lonza), 2 mM Glutamine (Lonza) and 1% Non Essential Amino Acids (Lonza), under 5% CO2 humidified atmosphere at 37 °C. Cell culture was performed accordingly to ISO 10993-5:2009 guidelines. The presence of mycoplasma in culture media was tested according to internal SOPs.

**Cytotoxicity assessment in vitro**. MRC5 cells were plated into 24-well plates. The NanoM1-BMP2 wound dressing was tested side-by-side with polyurethane film containing 0.1% zinc diethyldithiocarbamate (known for inducing cytotoxic effects; Hatano Research Institute/Food and Drug Safety Center, Japan) and high density polyethylene film (negative control; Hatano Research Institute). To assess cytotoxicity, pieces of different size (20, 16, 9, 4, 1 mm²) were placed in contact to the cultured cells, when 70–80% confluence was reached. Cells were cultured in the presence of the membranes for 3 days before being examined microscopically for changes in the general morphology, presence of vacuolization, detachment, lysis and membrane integrity, following the criteria for the qualitative evaluation of cytotoxicity according to ISO 10993 guidelines, part 5 (2009) and part 12 (2012): Class 0, no reactivity (no effects around or below sample); Class 1, slight reactivity (few malformed or degenerated cells); Class 2, mild reactivity (small area of malformed or degenerated cells below the sample); Class 3, moderate reactivity (malformed or degenerated cells in an area larger than the size of the sample but ≤1 cm²); Class 4, severe reactivity (malformed or degenerated cells in an area larger than the size of the sample but >1 cm²). A grade higher than 2 was considered as cytotoxic.

**Quantitative cell viability assay**. As a quantitative measure of cytotoxicity, cell viability was evaluated. At day 3, membranes were discarded, cells were washed twice with PBS, fed with 1 ml culture medium and 100 µl/well of Cell Viability Reagent WST-1 (Lonza) was added to each well, according to internal SOPs. The cells were incubated for 3 h at 37 °C in 5% CO₂, and 100 µl of supernatant were transferred into a 96-well plate. Absorbance was measured at 450 and 620 nm in a Multiskan EX device (Thermo Fisher Scientific, Graffenstaden, France). Data analysis was performed with Ascent 2.6 (Thermo Fisher Scientific). Results were expressed as percentage of viable cells in respect to a blank control. A decrease of 30% viability was considered as cytotoxic.

**Experiments with animals**. Animal experiments were performed according to the ethical guidelines for animal experiments[59]. The protocols used, included in the project "Toxicologie Réglementaire", was authorized by the "Ministère de l'Enseignement supérieur et de la Recherche" No. 01191.02. Seven-weeks-old rats were maintained for at least 5 days in Specific Pathogen Free rooms (authorized by the French Ministries of Agriculture and Research; agreement No. A35 288-1) before the beginning of the study, according to internal SOPs, under controlled conditions of temperature (22 ± 3 °C), humidity (50 ± 20%), photoperiod (12 h light/12 h dark) and air exchange, according to internal SOPs. Animals were housed in standard-size polycarbonate cages (with filter lid), and bedding was replaced twice a week.

**Intra-articular implant of ARTiCAR in nude rats**. Evaluation of acute toxicity in vivo was achieved via intra-articular implant of ARTiCAR in a model of induced osteochondral defect in RH-Foxn1 rnu/rnu nude rats (Harlan, Gannat, France). Briefly, sterile NanoM1-BMP2 were rinsed in sterile PBS and cut into quarters (1.77 mm²) before implant. Subconfluent human bone marrow MSCs (Promocell, Heidelberg, Germany) were washed and resuspended in hyaluronic acid/alginate mixture, as previously published[28,29], to a concentration of 3.0 × 10⁷ cells/ml. Prior to implant, rats were anesthetized with intraperitoneal injection of a solution of 70 mg/kg ketamine and 10 mg/kg xylazine. After shaving and disinfection of right hind leg, round 1.5 mm osteochondral defects were induced with a short drill in the patellar groove of the femur, in the midline of the femoral trochlea, until bleeding of the subchondral bone (approx. 2 mm). The NanoM1-BMP2 membrane was placed at the bottom of the defect, which was in turn filled with hMSCs/hydrogel mix and gelled via drop-wise addition of 102 mM calcium chloride (Sigma-Aldrich), over 5 min. These rats constituted experimental group 1 (ARTiCAR; n = 20 rats; 10 males and 10 females; 3.5 µl of hydrogel containing 105,000 ± 10% cells). Other rats were subject to the same procedure, but implanted with hydrogel only (as vehicle) and constituted group 2 (n = 20 rats; 10 males and 10 females; 3.5 µl of hydrogel). After gelation, the articulation capsule was closed, muscle and skin were sutured and the wound was thoroughly disinfected with povidone-iodine solution. After surgery, rats were kept under observation for post-anesthesia recovery. After recovery, 0.05–0.1 mg/kg buprenorphine was administered by subcutaneous injection. Animals were allowed unrestricted movement for the duration of the study (90 days). Rats were monitored daily for wound healing, leg mobility, morbidity, mortality and evident sign of toxicity.

**Blood analysis**. At day 7 post implant, five male and five female fasted rats/group (n = 20) were anesthetized with excess isoflurane and ventricular blood was collected either in EDTA-containing tubes or in heparin-containing tubes, for

hematological or biochemical analysis, respectively. Between day 7 and 90, the remaining rats were observed and monitored twice a week for any loss of weight. Haematocrit, haemoglobin concentration, erythrocyte count, leukocyte counts, mean corpuscular volume and platelet count were determined in the blood samples on the day of collection by impedance variation and photometry (MINDRAY BC 2800 hematology analyzer, 4M, France). For biochemistry evaluation, plasma samples were prepared according to internal SOPs. Sodium, potassium, chloride, calcium, inorganic phosphate, glucose, urea, creatinine, total bilirubin, total cholesterol, triglycerides, aspartate aminotransferase (ASAT), alanine aminotransferase (ALAT), total proteins, albumin, and albumin/globulin ratio were quantified (Cobas Mira biochemistry analyzer, 4M, France).

**Histopathology analysis of nude rats implants**. Histopathology analysis was conducted on the fasted animals used for blood test. Briefly, a macroscopic autopsy was performed on freshly euthanized rats. Organs (treated knee, spleen, mesenteric lymph nodes, liver, lungs with bronchi and bronchiole, kidneys and heart) were macroscopically observed, explanted and collected. The right hind paw was sectioned at the epiphyses of both femur and tibia to recover knee joints subject to implant. Spleen, liver, kidneys and heart were weighed and preserved with the other organs at room temperature in 4% formalin (Sigma-Aldrich) until histological analyses. Organs were fixed in 4% paraformaldehyde, dehydrated, embedded in paraffin, sectioned and examined for histopathology.

**Tissue harvesting for human DNA qPCR**. Ninety days post implant, five male and five female rats/group ($n = 20$) were euthanized by exsanguination under anesthesia. After a macroscopic autopsy organs (ovaries with oviducts, testes, brain, treated knee, spleen, liver, kidneys, lungs, bone marrow, heart and the skin covering the treated knee joint) were weighed and collected for DNA extraction using the NucleoBond AXG100 kit (Macherey Nagel, Hoerdt, France), following manufacturer's instructions. Briefly, all tissue samples except knee joints were homogenized in M-tubes (Miltenyi Biotec, Paris, France) containing buffer G2 on a GentleMACS Dissociator (Miltenyi Biotec, Paris, France). Knee joints were homogenized using Ultra-Turrax® dissociator instrument in buffer G2. Following extraction, the DNA pellet was dissolved in molecular biology grade water, and stored at 20 °C. Quantitative PCR (qPCR) with the iTaq Universal Probes Supermix (BioRad, Marnes-la-Coquette, France) was used to quantify human Alu sequences with the TaqMan AluYB8 Probe (Thermo Fischer Scientific). Genomic DNA from the different tissues of the implanted rats was amplified side-by-side with DNA from control rats, spiked-in with variable amount of DNA from U87-MG human cells (22, 7, 0.7 ng; 70, 7, 2.2, 0.7, 0.07 pg or no DNA), to build a standard curve. Samples were run in triplicate for 35 cycles on a CFX System (Bio-Rad). The limit of detection corresponded to the average signal from control rat DNA not spiked-in with human DNA.

**Harvesting of MSCs from sheep bone marrow**. Four weeks ($28 \pm 2$ days) prior to interarticular surgery, adult sheep females (Rideau Arcott Hybrids strain) were subject to bone marrow aspiration procedure. Briefly, animals were placed in ventral recumbency and anesthetized with a mix of glycopyrrolate, xylazine and ketamine administered intramuscularly (IM). An IV catheter was placed in the appropriate vein. The larynx was sprayed with lidocaine and the animals were intubated with an appropriate sized cuffed orotracheal tube. If intubation was not possible under IM anesthesia, induction was performed using isoflurane in $O_2$ (1–5%) or propofol intravenously. The sheep were then mechanically ventilated with isoflurane in $O_2$. The harvest site was disinfected and a needle was introduced in the iliac crest. A sterile 10 mL syringe was filled with 1 mL of 5000 IU/mL heparin and filled with ~8 mL of bone marrow. The syringe containing bone marrow sample and heparin was sealed with an appropriate sterile cap for mesenchymal stem cells isolation, characterization and preparation for the surgical procedure.

**Isolation and expansion of bone marrow-derived sheep MSCs**. The MSCs harvested from sheep iliac crest were isolated according to their adherence to cell culture plastic. Bone marrow aspirates were first washed by addition of an equal volume of phosphate buffer saline (PBS; Sigma-Aldrich, France) and centrifuged at $220 \times g$ for 5 min. The cell pellets were suspended in Dulbecco's Modified Eagle Medium (DMEM; Lonza, Germany) containing 10% heat-inactivated fetal bovine serum (Gibco, Thermo Fisher Scientific, France), 50 U/mL of penicillin (Lonza, Germany), 50 µg/mL of streptomycin (Lonza, Germany), 2.5 µg/mL Fungizone (Lonza, Germany), and seeded in a T75 culture flasks, under standard cell culture conditions. The following day, medium was discarded and attached cells were gently washed up several times with PBS to remove non-adherent cells. Flasks were then incubated for several days in DMEM, replaced every 72 h to promote emergence of colonies from adherent cells. When cells finally reached sub-confluence, they were sub-cultured until passage 2, when they were expanded for stemness characterization.

**Characterization of sheep MSCs**. The MSCs were characterized according to their ability to form colonies and to their multipotency. Colony-forming unit-fibroblast (CFU-F) assays were performed in triplicates ($n = 3$) with two different

ranges of serial dilutions. After 14 days of culture, MSCs were rinsed with PBS, and fixed with 4% (w/v) paraformaldehyde. The colonies were stained using hematoxylin/eosin (Sigma-Aldrich, France) and counted. The potential of isolated sheep cells to undergo trilineage differentiation was demonstrated in vitro. For adipogenic differentiation, sheep cells were seeded at $2.1 \times 10^4$ cells/cm$^2$ and cultured for 3 days with proliferation culture medium. After that, they were induced to differentiate by means of 3 alternate adipogenic induction/maintenance cycles. For each cycle, cells were cultured for 3 days in standard induction medium (Lonza, PT-3004), followed by 3 days of culture in standard maintenance medium (Lonza, PT-3004). After three cycles, cells were then cultured for 7 days with maintenance medium, replaced every 72 h. Negative controls for adipogenic differentiation were cultured in DMEM. After 28 days of culture, cells were rinsed with PBS, fixed in 4% (w/v) paraformaldehyde, rinsed with 60% (v/v) isopropanol and finally stained with 0.5% (w/v) Oil Red O solution to detect lipid vesicles. For chondrogenic differentiation, sheep cells were seeded as a pellet at the density of $2.5 \times 10^4$ cells and cultured in chondrogenic medium (Lonza, PT-3925) supplemented with 10 ng/mL of TGF-β3 growth factor (Peprotech, France). Negative control for chondrogenic differentiation were cultured in DMEM. After 28 days of culture, pellets were rinsed with PBS, fixed in 4% (w/v) paraformaldehyde, and embedded in paraffin. Paraffin sections were stained with Alcian Blue solution (Sigma-Aldrich, France) to visualize glycosaminoglycans and with Fast Red solution (Sigma-Aldrich, France) to visualize cell nuclei. For osteogenic differentiation, sheep cells were seeded at $2.1 \times 10^4$ cells/cm$^2$ and cultured for 3 days in DMEM. After, they were cultured in osteogenic medium (Lonza, PT-3002). Negative controls for the osteogenic differentiation were cultured in DMEM. After 21 days of culture, cells were rinsed with PBS, fixed with ice-cold 70% (v/v) ethanol and stained with Alizarin Red solution (40 mM, pH 4.1; Sigma-Aldrich, France) to detect calcium deposits.

**Immunophenotypic characterization of sheep MSCs**. For cell surface markers analyses, cells were treated with 0.5% (w/v) bovine serum albumin and double-stained using monoclonal antibodies, including phycoerythrin (PE)-conjugated mouse anti-human CD34 (BD Biosciences, Le-Pont-de-Claix, France) with fluorescein isothiocyanate (FITC)-conjugated mouse anti-human CD90 (Beckman Coulter, Villepinte, France), PE-conjugated mouse anti-human CD166 (Beckman Coulter, Villepinte, France) with FITC-conjugated mouse anti-human CD45 (Dako, Glostrup, Denmark), PE-conjugated mouse anti-human CD105 (Beckman Coulter, Villepinte, France) with FITC-conjugated mouse anti-human CD44 (Beckman Coulter, Villepinte, France), and PE-conjugated mouse anti-human CD73 (BD Biosciences, Le-Pont-de-Claix, France) with FITC-conjugated mouse anti-human leukocyte antigen-antigen D related (HLA-DR) (Beckman Coulter, Villepinte, France). Antibodies were used at a dilution of 1:10 (PE CD34, FITC CD90, PE CD166, FITC CD44, PE CD105, FITC HLA-DR and PE CD73) or 1:20 (FITC CD45). Appropriate isotype-matched control antibodies named FITC or PE mouse IgG1 (Dako, Glostrup, Denmark) were used in each analysis. Cells were then examined by flow cytometry using a BD LSR II flow cytometer (Becton Dickinson Biosciences, San Jose, California). Fluorescence intensity and percentage of antigen positive cells were determined for each surface marker.

**Induction of osteochondral defect in sheep**. A total of 16 adult sheep underwent surgical induction of osteochondral defect into the medial femoral condyle. Three groups of sheep (ARTiCAR, AG control, NT control) were considered; each sheep was implanted on either the proximal or distal part of the right or left condyle of posterior legs (supplemental table 1). For surgery, the hind limb was flexed to a position at which the medial condyle could be palpated under the skin. A 15 cm medial parapatellar skin incision was performed. After blunt dissection of the subcutaneous tissues, the fascia overlying the *vastus medialis* muscle was incised just distal to the belly muscle with a small incision parallel to the muscle fibers and the *vastus* was retracted proximally. Blunt dissection was used to expose the periosteum down to the medial condyle of the femur. The joint capsule and periosteum were incised just proximal to the origin of the medial collateral ligament. Overlying soft tissues were removed from the bone only in the vicinity of the drill holes. Holes were predrilled using a 6-mm drill bit to a depth of 3 mm, except for the AG group, where the hole had a depth of 6 mm.

**Intra-articular implant of ARTiCAR in sheep**. Following the induction of the defect, the NanoM1-BMP2 was placed, and the defect was filled with MSCs/hydrogel mix (ARTiCAR combined ATMPs, $n = 9$). In the AG group ($n = 10$), a bone sample of 6 mm of diameter and 6 mm deep was taken out from the condyle and placed into the defect. In the NT group ($n = 13$) the defect was neither treated, nor filled. Up to 5 mL 0.25% bupivacaine were infiltrated into the surgical site to achieve local anesthesia and manage pain after surgery. The tissues were repositioned and closed layer-by-layer using appropriate sutures. Postoperative analgesia and antibiotic therapy were performed, 5 mg/kg Excede (IM) was administered during recovery from anesthesia, and 4 mg/kg Carprofen (IM) was administered 3 days after surgery.

**Non-invasive monitoring of ARTiCAR via MRI**. For the longitudinal analysis of the knee repair, sheep were examined three times via MRI, immediately after surgery, at 15 and 26 weeks, using a Magnetom Verio 3 T (Siemens). For the

procedure, sheep were anesthetized with an intravenous injection of 0.05 mg/kg xylazine and 5 mg/Kg ketamine and placed in dorsal decubitus. A total of six sites of surgery were imaged for each group. Proton density-weighted, fat-saturated sagittal sections of the acquisitions were analyzed using the Osirix open-source software.

**Three-dimensional micro-CT of explanted femoral condyles**. For analysis of the bone mineralization, sheep were anesthetized, weighed and euthanized by a lethal injection of 540 mg/ml Euthanyl rapid IV bolus 26 weeks after surgery. Death was confirmed and recorded by observation of asystole or ventricular fibrillation, either on the electrocardiogram or by auscultation. Femoral condyle from were explanted from euthanized sheep and imaged via 3D micro-CT (Quantum Fx mCT, Julien Becker, ICS, IGBMC, Strasbourg, France). A total of six sites of surgery were imaged for each group. Three-dimensional surface rendering was obtained from micro-CT 2D images using the Osirix open-source software.

**Histopathology analysis of implants in sheep**. Treated femurs were removed from euthanized animals and subject to macroscopic inspection of the articular surface. The distal femoral epiphysis (with condyles) were individually identified and collected in 10% neutral buffered formalin, after macroscopic examination. Bone blocks were cut in two halves, by sawing in the middle of the sample along its longitudinal axis. Sections were cut through the defect along its deeper axis, from the bone surface to the end of the drill hole producing rectangular-shaped defect half sections. Full-thickness femoral bone-cartilage defect sites underwent undecalcified bone preparation and were infiltrated with methyl methacrylate and polymerized. A single 8 μm section spanning the entire width of the defect was cut along the parasagittal plane from each medial femoral condyle. The sections were stained with safranin o–fast green, for the staining of both cartilage and bone. The femoral defect sites were carefully evaluated and scored according to the ICRS histological score system[39] (Supplemental table 2). Also, the tissue underneath and adjacent to the defect was evaluated for a number of parameters (Supplemental table 3) to assess safety and efficacy of the treatments.

**Statistical analysis**. Results from the WST1 assay were statistically evaluated using one-way ANOVA followed by Tukey post hoc test on Prism 4.03 (GraphPad). A $p$ value ≤ 0.05 was considered significant. One-way ANOVA followed by Bonferroni post-hoc test was used to compare hematological and biochemical parameters in the blood tests, using Prism 4.03. A $p$ value ≤ 0.05 was considered significant. Both SigmaPlot (SYSTAT Software) and Prism 5.0 (GraphPad) were used to compare the ICSR II scores from the in vivo experiments in sheep. Equal variance test and normality tests were performed. Either one- (differences induced by treatment) or two-way ANOVA (differences induced by both treatment and time) followed by Bonferroni post hoc test were used to assess significant differences among the continuous variables of the study groups. If either equal variance test or normality test failed, a Kruskal–Wallis one-way ANOVA with Dunn's correction was conducted. A $p$ value ≤ 0.1 was considered significant.

**Reporting summary**. Further information on research design is available in the Nature Research Reporting Summary linked to this article.

## Data availability

All the original data discussed in this work are available upon request

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

## Acknowledgements
This work was supported by the ARTiCAR ANR grant (Agence Nationale de la Recherche) and by SATT (Société d'Accélération du Transfert de Technologies) Con-ectus Alsace. Authors are thankful to "Faculté de Chirurgie Dentaire de Strasbourg" for the financial support. We are grateful to Stéphanie Krissian for sheep surgery and Hans Adriaensen for MRI acquisitions (CIRE Platform for experimental surgery and imaging, Val de Loire INRA Centre, 37380 Nouzilly, France).

## Author contributions
L.K., L.P. and Y.I.-G. prepared and characterized the active implants, contributed to the experiments and to the analyses of data; L.P. and F.B. performed and analyzed MRI; L.B. and M.T. prepared MSCs and performed qPCR; L.K., L.P. and L.G. contributed to the analyses of the data and drafted the manuscript. P.A. designed acute cytotoxicity and biodistribution experiments, and acquired data; R.M.G.-D. and E.G.B. contributed to the study design and supervision; N.B.-J. conducted the hypothesis, designed the experiments, directed the work and contributed to draft the manuscript.

## Additional information

**Competing interests:** The authors declare no competing interests.

