## [Peer Review File · Nature Communications]

Reviewers' Comments:

Reviewer #1:

Remarks to the Author:

Comments to authors

In this work, Keller and coauthors performed investigations aimed at testing a novel graft (ARTiCAR: ARTicular Cartilage and subchondral bone implant, combination of a wound dressing made of nanofibrous PCL nano-functionalized with BMP-2 [for subchondral bone repair] and autologous mesenchymal stromal cells embedded in a hyaluronic acid/alginate-based hydrogel [for cartilage repair]) for the feasibility and safety in treating osteochondral grafts. In particular the authors performed several experiments in accordance to the international regulatory guidelines for cell therapies and medical devices and provided evidences for the safe use of ARTiCAR for cartilage repair. However, the study suffers from these main weaknesses:

1. ARTiCAR is a ATMP (advanced therapy medicinal products) that contains ex vivo cultured autologous bone marrow derived mesenchymal stromal cells (MSCs). The authors should have performed the study in nude rats, using MSCs isolated and cultured with the specific protocols envisioned for this ATMP, instead of MSCs from PromoCell (i.e., cells isolated expanded and cryopreserved with protocols that are PromoCell's proprietary). Considering that the nature and the behavior of MSCs can substantially differ dependly to the protocols used for their isolation from the bone marrow sample, and their subsequent culture, the reported results from this study are valid only for the product generated using the PromoCell and can not be extended to the real ATMP (that, instead, would contain MSCs cultured under different conditions).

2. Tumorigenicity: The authors wrote: "... since tumorigenicity of MSCs is debated, the biodistribution of hMSCs is a critical concern of preclinical safety" (Page 7, lines 20-21). However, in order to fully address this safety issue, not only biodistribution, but also the tumorigenicity of the transplanted cells must be assessed (as reported in this study: Zscharnack et al. Journal of Translational Medicine (2015) 13:160; DOI 10.1186/s12967-015-0517-x). Indeed it can be that the MSCs are not capable to migrate from the graft site but capable to form or induce the formation of tumor.

Additional points

3. Scale bars in Figure 2 are missing

4. Descriptions of what the asterisks and arrows in Figure 4A,B and C show, are missing

5. Several different names are used to distinguish the groups in the large animal study: "CTR+" or "autograft (AG)"; "CTR-" or "defect" or "no-treatment control (NT)" (example: Figure 4 vs Supplemental table 1. It would be better, instead, to use always a single definition for the considered groups through the text.

6. Page 7, line 5-6. The authors wrote that "total knee arthroplasty (TKA) "is suboptimal in young, ...as it induces fibrocartilage formation, cellular hyper- or hypertrophy and lack of proper interface between cartilage and subchondral bone". This is not correct! The mentioned limitations are characteristic to other cartilage repair strategies. The main issues associated to TKA, instead are others, e.g.: risk of infections, risk of joint dislocation, implant wear and loosening (and consequently need of revision procedures to replace the original components). The aforementioned sentence must be corrected.

7. Page 7, lines10-11. The authors wrote: "Similarly to other smart implantable scaffolds that promote osteochondral differentiation 40, 41, 42, the ARTiCAR releases BMP-2". However no one of the "smart implantable scaffolds" described in the cited studies releases BMP-2! This sentence must be rephrased, or different studies must be cited.

8. Page 7, lines 27-28. The authors wrote that ARTiCAR "can enter phase I clinical trials as a treatment for OA, tendon degeneration and other age-related degenerative musculoskeletal issues". This is not correct: the proposed ATMP can be considered for the treatment of cartilage defects but not for the other aforementioned issues! The authors must correct this sentence. Moreover they have to state which are the specific indications that they would like to target with ARTiCAR.

9. Page 7. Considering that ARTiCAR incorporates MSCs, it is required that in the discussion section, the authors mention the quality controls and the release criteria that they are intended to

implement on the MSCs prior to their therapeutic application.

10. Page 12, lines 18-19. The authors wrote: "The syringe containing bone marrow sample ...for mesenchymal stem cells isolation, characterization and ...". Which specific characterization was performed on the MSCs? In addition it is important to add some details of the cell protocol used to culture the cells: (i) expansion condition (source of the serum used [autologous or bovine?], presence or absence of growth factors in the expansion medium, ...), (ii) extent of MSCs expansion [cell population doublings and/or number of passaging]).

Reviewer #2:

Remarks to the Author:

The abstract is too general and does not give enough information about the paper. Explain what you do and avoid commercial names. Structure or order your abstract and give your concrete results. Try to focus your introduction with your purpose. Explain what is your proposal and what you want to improve. Explain the material you have used and, again, avoid commercial names.

You say, "Recently, membrane collagen ...". Is true, the cites are recently, but the system is old enough.

If you have many correlative cites, write the first and the last separated by a script. In the introduction explain your proposal, write the former attempts, and finish with your hypothesis and objectives

After the introduction introduce your Material and methods section. You included it at the end of the paper.

Why you combined mice and sheep in the same paper? In my opinion there are two different models. Is better if you write two separate papers.

Bibliography is too general and many cites have no relations with your paper. Cites related with growth factors in cartilage repair should be included

The discussion is poor, try to focus with your results.

Papers including bone ceramics and growth factors have been published in the last years and you do not cite them.

In the introduction you cite 36 papers!!! in the discussion only 20, of those 10 new citations

A clear conclusion must be given

I do not understand the advantages of your product in relation with former commercial products. Why should we use your proposal?

Reviewer #3:

Remarks to the Author:

The work by Nadia Benkirane-Jessel et al. carried out animal studies in an osteochondral model with their combined biomaterial/stem cell system. This work is on the basis of their previous studies on nanofibers and nanofiber-BMP delivery for bone (as referenced 21-24), and hydrogel system that can incorporate cells for cartilage (as referenced 25-27). Although the authors aim to reap up the merits of two different systems (one is gene delivery for bone while the other is stem cell delivery) in one pot (osteochondral), the combined system lacks any novelty in terms of biomaterials design and methodology. Although they highlight the current focus on the preclinical safety issue of their old versions, the findings in this study are considered only at the marginal from a scientific point-of-view, failing to make significant technological advances or providing scientific insight in the design of biomaterial-based tissue engineering. The animal models they performed are also well-known in this community, failing to draw significant attention. This work is more specified to prove the preclinical performance of what they have been developed, thus is more suitable to the application-targeted or biomaterials-related journals.

Point-by-point response to Reviewers' comments

Reviewer #1 (Remarks to the Author):

We would like to thank the Reviewer for the positive opinion concerning the interest and the potential impact of our manuscript, and for the constructive remarks that were important in improving the quality of the manuscript.

The regeneration of the cartilage and of the subchondral bone is a big unsolved issue. Scientists are on the constant search for new technologies that could help bridging the gap towards better translational solutions. We agree that the approach that we propose in the submitted manuscript is not new *tout-court*. However, it takes into account the complexity of the articular regeneration, as the currently available solutions are far too simplistic to solve the issue. Therefore, we believe that the novelty of our message is that the ARTiCAR has the potential to address osteochondral defects better than currently available solutions, as following the implant in large mammals, it achieved higher vascularization and lower fibrocartilage formation than the autograft control in areas immediately outside of the induced bone defect. Moreover, ARTiCAR-mediated OAR could be assessed non-invasively, by means of MRI. Above all, the ARTiCAR possess the safety requirements to undergo human tests, having shown neither sign of toxicity, nor concerns for tumor formation of MSC derivation. Altogether, in trying to provide a complex solution for a complex problem, we submitted data that convincingly speak in favour of the safety of the ARTiCAR, which are an indispensable requisite for proceeding to the clinical trial phases.

Comments to authors

In this work, Keller and coauthors performed investigations aimed at testing a novel graft (ARTiCAR: ARTicular Cartilage and subchondral bone implant, combination of a wound dressing made of nanofibrous PCL nano-functionalized with BMP-2 [for subchondral bone repair] and autologous mesenchymal stromal cells embedded in a hyaluronic acid/alginate-based hydrogel [for cartilage repair]) for the feasibility and safety in treating osteochondral grafts. In particular the authors performed several experiments in accordance to the international regulatory guidelines for cell therapies and medical devices and provided evidences for the safe use of ARTiCAR for cartilage repair. However, the study suffers from these main weaknesses:

1. ARTiCAR is a ATMP (advanced therapy medicinal products) that contains ex vivo cultured autologous bone marrow derived mesenchymal stromal cells (MSCs). The authors should have performed the study in nude rats, using MSCs isolated and cultured with the specific protocols envisioned for this ATMP, instead of MSCs from PromoCell (i.e., cells isolated expanded and cryopreserved with protocols that are PromoCell's proprietary). Considering that the nature and the behavior of MSCs can substantially differ dependly to the protocols used for their isolation from the bone marrow sample, and their subsequent culture, the reported results from this study are valid only for the product generated using the PromoCell and can not be extended to the real ATMP (that, instead, would contain MSCs cultured under different conditions).

The reviewer is right in saying that the behaviour of MSCs largely depend on the methodology used for harvesting and culturing them; however, although the MSCs transplanted in nude rats were from a commercial source, the MSCs transplanted in sheep were from the sheep, and were collected, expanded and characterized AS THEY WERE "real" human ATMPs. To further address the Reviewer's concerns, we would like to emphasize that 1) the Investigational Medicinal Product Dossier (IMPD), 2) the Investigator Brochure (IB) of the combined ATMP, as well as 3) the Clinical Trial protocol already exist as draft versions, which only need to be amended, as the preclinical safety phase 1 study has been validated by C.Ris Pharma and the efficacy has been demonstrated. Authorization by ANSM will be granted as soon as the relevant protocol is used to harvest, characterize and expand patient-specific MSCs (MSCs as ATMP, Ecellfrance CTSA Clamart, ANSM authorization ETI/14/O/006). Although such a step is needed for the clinical trial phase 1, it is not needed for preclinical trial phase 1, presented in this manuscript.

In conclusion, we used two different types of MSCs, from two unrelated sources (human and sheep) and likely (although we cannot know PromoCell proprietary protocol) two different methodologies. We believe that the results presented in such a way are solid and fulfil the requirements of the safety assessment.

2. Tumorigenicity: The authors wrote: "... since tumorigenicity of MSCs is debated, the biodistribution of hMSCs is a critical concern of preclinical safety" (Page 7, lines 20-21). However, in order to fully address this safety issue, not only biodistribution, but also the tumorigenicity of the transplanted cells must be assessed (as reported in this study: Zscharnack et al. Journal of Translational Medicine (2015) 13:160; DOI 10.1186/s12967-015-0517-x). Indeed it can be that the MSCs are not capable to migrate from the graft site but capable to form or induce the formation of tumor.

Thanks for the opportunity to stress the relevance of assessing the tumorigenicity of the transplanted MSCs. Although such an assessment is mandatory for phase 1 clinical trial, it is not for the phase 1 preclinical study presented in this manuscript. Nonetheless, we would like to stress that the biodistribution of the MSCs was performed body-wide. Since no MSCs DNA could be amplified above the threshold level in all the nude rats transplanted with ARTiCAR, then we concluded that no MSCs —or tumors formed from them— could be found in the transplanted rats and the experiment's endpoint.

Moreover, in the experiments with sheep, autologous MSCs were used, thoroughly characterized and expanded, as now described in the methods section of our manuscript. The sheep used for the experiments were monitored for either 12 or 26 weeks, and no sign of tumorigenicity was detected, neither at the transplantation site, nor elsewhere. Together, these data suggest that, in our experimental conditions, MSCs in the ARTiCAR combined ATMPs do not give raise to concerns for tumorigenesis.

Additional points

3. Scale bars in Figure 2 are missing

4. Descriptions of what the asterisks and arrows in Figure 4A,B and C show, are missing

5. Several different names are used to distinguish the groups in the large animal study: "CTR+" or "autograft (AG)"; "CTR-" or "defect" or "no-treatment control (NT)" (example: Figure 4 vs Supplemental table 1. It would be better, instead, to use always a single definition for the considered groups through the text.

6. Page 7, line 5-6. The authors wrote that "total knee arthroplasty (TKA) "is suboptimal in young, ... as it induces fibrocartilage formation, cellular hyper- or hypotrophy and lack of proper interface between cartilage and subchondral bone". This is not correct! The mentioned limitations are characteristic to other cartilage repair strategies. The main issues associated to TKA, instead are others, e.g.: risk of infections, risk of joint dislocation, implant wear and loosening (and consequently need of revision procedures to replace the original components). The aforementioned sentence must be corrected.

7. Page 7, lines 10-11. The authors wrote: "Similarly to other smart implantable scaffolds that promote osteochondral differentiation 40, 41, 42, the ARTiCAR releases BMP-2". However no one of the "smart implantable scaffolds" described in the cited studies releases BMP-2! This sentence must be rephrased, or different studies must be cited.

8. Page 7, lines 27-28. The authors wrote that ARTiCAR "can enter phase I clinical trials as a treatment for OA, tendon degeneration and other age-related degenerative musculoskeletal issues". This is not correct: the proposed ATMP can be considered for the treatment of cartilage defects but not for the other aforementioned issues! The authors must correct this sentence. Moreover they have to state which are the specific indications that they would like to target with ARTiCAR.

Points 3-8 were addressed according to the Reviewer's suggestions

9. Page 7. Considering that ARTiCAR incorporates MSCs, it is required that in the discussion section, the authors mention the quality controls and the release criteria that they are intended to implement on the MSCs prior to their therapeutic application.

A sentence has been added to the discussion section concerning the quality control suggested for hMSCs to be used in future clinical trials, as well as the minimum release criteria adopted. A table (suppl. table 4) summarises both the implemented QC (at seeding, i.e. quality of the bone marrow aspirate and at harvesting, i.e. quality of the hMSCs to be transplanted) and the release criteria, according to Dominici et al., *Cytotherapy*, 2006. Briefly, hMSC will be manufactured according to GMP rules using aseptic procedures and disposable sterile single-use supplies for all product contact steps. The cell culture is performed according to the Cell Production Unit (CPU-HUPHM, Madrid, Spain) quality system, with established standard operation procedures (SOPs) and methods. This Unit has a wide experience in ATMP production and clinical trials since 2005 and has obtained the mandatory GMP certificate since 2010.

Briefly, the whole process will be conducted in accordance with written procedures and each step is recorded on batch records. All manipulations involving the initial preparation of cells, cell culture and cell packaging are performed in clean rooms of appropriate class of air cleanliness. All manufacturing staff is trained in use of the process pertinent SOPs including the line clearance and disinfection procedures. The batch production records for each lot require documentation and confirmation signatures that the procedures have been followed.

Throughout the whole process, a tracking, labeling and verification system ensures that the patient receives the correct product (autologous MSCs).

The quality controls we intend to implement for the GMP grade human MSC for clinical application are the following:

1. For initial bone marrow (BM) sample quality controls (at seeding day): cell count, viability of BM, microbial sterility (including aerobic and anaerobic micro-organisms), Mycoplasma test and clonogenicity assay (CFU-f).
2. For MSC quality controls of final product (at harvesting day): cell count, viability, immunophenotype (including: HLA I, CD 73, CD 90, CD 105, CD166, HLA-DR, CD 31, CD 34, CD 45 and CD 80), microbial sterility (including aerobic and anaerobic micro-organisms), Mycoplasma test, endotoxin test and clonogenicity assay (CFU-f).

The release criteria of the final MSC product will include all mandatory specifications for this type of ATMP and will be: 1) a cell count $> 24 \times 10^6$ MSC (if for example, target dose is 30×10^6 MSC); 2) a viability $> 90 \%$; 3) an immunophenotype with $\geq 90 \%$ MSCs positive for CD 73, CD 90, CD 105, and CD166 and $\leq 10 \%$ MSCs positive for HLA-DR, CD 31, CD 34, CD 45 and CD 80; 4) a morphology and aspect as follows: fibroblasts with spindle shape and white suspension without aggregates (Dominici et al., 2006).

10. Page 12, lines 18-19. The authors wrote: "The syringe containing bone marrow sample ...for mesenchymal stem cells isolation, characterization and ...". Which specific characterization was performed on the MSCs? In addition it is important to add some details of the cell protocol used to culture the cells: (i) expansion condition (source of the serum used [autologous or bovine?], presence or absence of growth factors in the expansion medium, ...), (ii) extent of MSCs expansion [cell population doublings and/or number of passaging]).

Detailed isolation, expansion and characterization protocols for sheep MSCs are now part of the methodology section.

Reviewer #2 (Remarks to the Author):

We thank Reviewer #2 for the comments.

The abstract is too general and does not give enough information about the paper. Explain what you do and avoid commercial names. Structure or order your abstract and give your concrete results

The abstract has a limited number of words, and has to be general, because it needs to provide understandable information to a broad —not a specialized— readership. Given the above considerations, we streamlined the text so that the abstract is now better structured and clearly shows the concrete results. As a side note, there are no commercial names in the abstract, only the acronym ARTiCAR

Try to focus your introduction with your purpose. Explain what is your proposal and what you want to improve. Explain the material you have used and, again, avoid commercial names.

You say, "Recently, membrane collagen ...". Is true, the cites are recent, but the system is old enough.

If you have many correlative cites, write the first and the last separated by a script. In the introduction explain your proposal, write the former attempts, and finish with your hypothesis and objectives

We agree with the Reviewer that the message that we passed in the introduction was not complete. Thanks to the suggestion, we have now streamlined the introduction so that the critical points mentioned by the Reviewer are clearer. The introduction gives first the rationale/problems, then goes through the state-of-the-art biotechnologies/biomedical techniques and eventually it briefly revises the main results of our manuscript, stressing the potential of the ARTiCAR for human use, since its safety was shown.

Concerning the literature quoted, we decided to give relevance to recent papers, even if written about old methods or technologies. There are countless articles on the topic, but we could not cite them all (the journal allows up to 70 references). The specific article mentioned by the Reviewer was quoted because a recently "refurbished" old technique was shown to fall short in promoting OAR, stressing once more the need for alternative approaches.

After the introduction introduce your Material and methods section. You included it at the end of the paper.

For our manuscript, we followed the author's guidelines of Nature Communications meticulously. The structure of the main text was organized according to the indications given under: <https://www.nature.com/ncomms/submit/article>

Why you combined mice and sheep in the same paper? In my opinion there are two different models. Is better if you write two separate papers.

We used cells in vitro and 2 animal models (rats and sheep) to evaluate the different aspects of the safety of the ARTiCAR at our best. In our opinion, this is an added value to the manuscript, rather than a defect. Moreover, one of the animal models used, the sheep, is considered a gold standard for orthopaedics.

Bibliography is too general and many cites have no relations with your paper. Cites related with growth factors in cartilage repair should be included

The ARTiCAR is a combined ATMPs of which we wanted to evaluate the feasibility, traceability and safety upon transplantation into an osteochondral defect in a large animal model. The efficacy of different growth factors tested was previously published by our group, and so it was for both the nanofunctionalized fibers and the MSCs (Mendoza-Palomares et al., ACS Nano, 2012; Eap et al., Biomed Mater Eng, 2012; Eap et al., Nano LIFE, 2014, Eap et al., Int J Nanomedicine, 2015; Keller et al., J Stem Cell Res Therapeutics, 2015; Keller et al., Nanomedicine, 2015; Keller et al., Int J Nanomedicine, 2017). In those articles, the relevant literature was revised thoroughly. The goal of this manuscript was to show the safety of the ARTiCAR, which is a cornerstone for downstream clinical trials to begin.

The discussion is poor, try to focus with your results.

Papers including bone ceramics and growth factors have been published in the last years and you no cite it.

In the introduction you cite 36 papers!!! in the discussion only 20, of those 10 news citations

A clear conclusion must be given

I no undestand the advantages of your product in relation with former commercial products. Why I should use your proposal?

We modified the discussion so that it is now more focused. We give relevance to aspects like the consequences of osteoarthritis and other osteochondral problems to both the life quality of patients and the health systems. We stressed that methodologies currently used have drawbacks that the ARTiCAR might overcome, and therefore we aimed to prove the safety of the RTiCAR, in order to proceed to clinical trials. The potential benefits of the ARTiCAR over other systems has been shown in previously published papers; the clear conclusion of the current manuscript is that the ARTiCAR is safe for human testing, with the potential to replace invasive techniques and achieve osteochondral regeneration without the need of prosthetic implants.

Reviewer #3 (Remarks to the Author):

We thank Reviewer #3 for the comments.

The work by Nadia Benkirane-Jessel et al. carried out animal studies in osteochondral model with their combined biomaterial/stem cell system. This work is on the basis of their previous studies on nanofibers and nanofiber-BMP delivery for bone (as referenced 21-24), and hydrogel system that can incorporate cells for cartilage (as referenced 25-27). Although the authors aim to reap up the merits of two different systems (one is gene delivery for bone while the other is stem cell delivery) in one pot (osteochondral), the combined system lacks any novelty in terms of biomaterials design and methodology.

Although they highlight the current focus on the preclinical safety issue of their old versions, the findings in this study are considered only at the marginal from scientific point-of-view, failing to making significant technological advances or providing scientific insight in design of biomaterial-based tissue engineering.

The submitted manuscript was meant to assess the safety of the ARTiCAR combined ATMPs, as a prerequisite to begin the clinical trials. Given the drawbacks of the methodologies currently used to treat osteoarthritis, we believe that addressing the safety issues of the ARTiCAR the way we did, providing plenty of data on three different biological systems (cells in vitro and two animal models) represents a substantial novelty and is of general interest to the broad readership of Nature Communications. Maybe not in terms of biomaterials (which is why we did not submit to a specialized journal), but definitely in terms of methodology (by “combining in one pot” different methodologies for achieving both cartilage and subchondral bone regeneration) and surely in terms of advancement in the experimental *iter* of ARTiCAR, which can now move on issues concerning the clinical trials.

Truth be told, we disagree with the Reviewer on the simplistic way it envisions biomedical research. In biology and medicine, 1+1 does not necessarily give 2. The combination of two or more previously reported methods/devices/therapeutics must be proved, in terms of feasibility, in terms of results achieved and, above all, in terms of safety.

The animal models they performed are also well-known in this community, failing to draw significant attention. This work is more specified to prove the preclinical performance of what they have been developed, thus is more suitable to the application-targeted or biomaterials-related journals.

In phase 1 preclinical studies like ours, where the safety of the proposed treatment to human is under question, it is seminal to use consolidated animal models. Sheep, in particular, is the gold standard in orthopaedics, therefore it is probably the most suitable animal models that we could use.

Because we did NOT put the focus on a novel biomaterial or on a novel biotechnology, but on the feasibility, traceability and safety of a combination of biomaterials AND therapeutics already known/used, we believe that the message passed by our manuscript fits better in Nature Communications than in any other application-targeted or biomaterials-related journals.

Reviewers' Comments:

Reviewer #1:

Remarks to the Author:

After a cautious checking of the authors' modifications and rebuttals to my raised points, I can state that Keller et al. have carefully addressing all my critiques in exhaustive and a detailed manner. Therefore, I don't have additional remarks.

Reviewer #2:

None

Reviewer #3:

None